# A comparison of methods to compute the rate of horizontal geomagnetic field variation

Samuel A. Fielding[1], Philip W. Livermore[3], Ciarán D. Beggan[2], Kathryn A. Whaler[1], and Gemma S. Richardson[2]

[1]School of GeoSciences, University of Edinburgh, Grant Institute, James Hutton Rd, King's Buildings, Edinburgh, United Kingdom EH9 3FE
[2]British Geological Survey, Lyell Centre, Research Avenue South, Edinburgh, United Kingdom EH14 4AP
[3]School of Earth and Environment, University of Leeds, Leeds, United Kingdom LS2 9JT

**Correspondence:** Samuel A. Fielding (sfieldin@ed.ac.uk)

**Abstract.** The rate of change of the horizontal external magnetic field is often used as a proxy for space weather activity and in particular for estimating geomagnetically induced currents (GICs) in high voltage power grids. This paper compares two commonly adopted methods for computing this rate of change: (1) the difference between consecutive measurements in the magnitude of the horizontal magnetic field, $H'$, and (2) the combined difference in the magnitude in the northward and eastward directions, usually denoted $R$. We find that there can be an absolute difference in the calculations between the two methodologies exceeding 100 nT/min during storm times for observatories in the sub-auroral zone, demonstrating that the choice between $R$ and $H'$ can make a significant difference to any GIC estimate. We also note an observable difference between the two methodologies during quiet times when the measurements are made close to the agonic line, though this difference does not have a significant impact on the efficacy of either of the two methodologies for GIC studies. Future studies should consider carefully the choice of geomagnetic indicators for estimation of GICs.

## 1 Introduction

Research into space weather and the effects from time-varying environmental conditions in the near-Earth environment has grown enormously in the past two decades. As the primary driver, the Sun generates physical phenomena such as, but not limited to, Coronal Mass Ejections which can strongly perturb the Earth's magnetic environment (Tsurutani et al., 1988) causing effects on modern technology (e.g. Mishra and Teriaca, 2023). Other potentially damaging space weather effects on technology come from solar flares (e.g. Grodji et al., 2022) and solar energetic particles (e.g. Reames, 1999; Usoskin and Kovaltsov, 2012; Tsurutani et al., 2024). At the boundary of the interplanetary magnetic field (IMF) and the geomagnetic field, termed the magnetopause, energy is transferred through to the Earth's magnetosphere, principally through the process of reconnection (Dungey, 1961; Tsurutani and Meng, 1972; Zhang et al., 2023; Dai et al., 2024). The response energizes the radiation belts (DeForest and McIlwain, 1971) and enhances ionospheric currents (Case et al., 2021), often causing a visible glow of the upper atmosphere known as the aurora (Zhou and Tsurutani, 2004; Gurram et al., 2021). Energy stored and released by the magnetotail produces bursts of geomagnetic activity, called substorms (Akasofu, 1964), which drive large rates

of change of the external magnetic field which in turn creates measurable variations in magnetic field measurements on the surface (Kanekal and Miyoshi, 2021).

Rapid geomagnetic disturbances induce a subsurface geoelectric field which, if sufficiently intense, can create Geomagnetically Induced Currents (GICs) in low resistance grounded technical infrastructure such as high voltage power grids (Pulkkinen et al., 2005) or gas pipelines (Tsurutani and Hajra, 2021). Geomagnetic disturbances are primarily driven by currents in the magnetosphere and ionosphere, especially the auroral electrojet at mid to high latitudes (Pulkkinen et al., 2017), though the equatorial electrojet and the ring current can also contribute to field variations at lower latitudes (de Villiers et al., 2017; Es-

pinosa et al., 2019), with the ring current contributing to GIC activity through its impact on region 2 field aligned currents (Bower et al., 2024). Both electrojets flow in the ionospheric E region (Alken and Egbert, 2025) while the ring current flows at a distance from the Earth of between 2 and 8 Earth radii (Tan et al., 2025). The auroral electrojets are mostly composed of Hall Currents (Guo et al., 2014) and are associated with eastward and westward electrojets that both flow from the dayside to nightside (Cowley, 2000). These electrojets are strongly associated with Field-aligned currents (Kamide and Akasofu, 1976)

and have strong coupling with solar wind parameters (Guo et al., 2014). These disturbances and their induction of GICs are most significant during strong geomagnetic activity (Despirak et al., 2024; Wawrzaszek et al., 2024), and the largest GICs are induced during the most active geomagnetic storm times such as the 2003 Halloween storms (Tsurutani and Hajra, 2021). Variation in the strength of the auroral electrojets primarily cause changes in the northward component of the horizontal geomagnetic field (Pirjola, 1998) whereas sudden storm commencements can induce significant changes in the eastward direction

as well (Smith et al., 2022b) and the equatorial electrojet can induce notable changes in both components (Kouassi et al., 2021). It is possible to interpret the magnetic field variation by modeling the electric field producing the magnetic field as two current systems, one externally-driven current system including ionospheric currents, and one internally induced in the shallow subsurface. The externally driven current system dominates in auroral areas, though the internal component can contribute up to 30% of a given perturbation (Juusola et al., 2020). In extreme storms, depending on local geology, geoelectric fields can

reach over 10 V/km in regions with high subsurface resistivity (Kelbert and Lucas, 2020).

    GICs are effectively quasi-DC in power grids and other critical infrastructure networks which can pose a significant economic hazard as they affect transformer operation, particularly through saturation of the hysteresis loop (Oughton et al., 2018; Rajput et al., 2021; Ramírez-Niño et al., 2016). Half-cycle saturation generates even harmonics, enhanced reactive power consumption and overheating which can cause permanent damage to critical infrastructure (Bolduc, 2002; Boteler, 2019; Abda et al., 2020).

Mac Manus et al. (2022) found that between 13% and 35% of transformers in New Zealand were at risk of damage through the impact of GICs. As GIC flow is not routinely monitored in many countries, methods for estimating it via proxy measurements have been developed (Marshall et al., 2011).

    The use of the ground-level magnetic field as an indicator of GIC activity relies on the time derivative of the geomagnetic field (Tsurutani and Lakhina, 2014), and more specifically of the 2D horizontal vector geomagnetic field $\mathbf{B}_H$, the projection

of $\mathbf{B}$ in the horizontal plane (Viljanen et al., 2001). As ground-level magnetic field is measured continuously at hundreds of dedicated observatories around the world, it can provide a proxy for regional GIC activity (Smith et al., 2021). Proxies for GIC are typically used as local measurements of GIC in electrical power grids are either non-existent or unavailable due to

the commercial sensitivity of the records. Where data are available, they often have restrictions on use (e.g. Rodger et al., 2017) which precludes release to the wider scientific community. Considering the time derivative of the 2D horizontal vector geomagnetic field rather than the total magnetic field is based on the induction response as used in magnetotelluric sounding. This establishes a frequency-dependent relationship between the magnetic to electric field variations as measured in the North ($\mathbf{B}_N$) and East ($\mathbf{B}_E$) directions (Robertson, 1987). $\mathbf{B}_N$ and ($\mathbf{B}_E$) are also often used as the sole magnetic field inputs to geoelectric field calculations (Wawrzaszek et al., 2023; Kelbert and Lucas, 2020; Wawrzaszek et al., 2024), though in theory the vertical component could be used to derive the divergence-free component of the geoelectric field (Vanhamäki et al., 2013). It is worth noting that although we only consider the 2D horizontal vector magnetic field, the vertical magnetic field can also be perturbed significantly during active times (Lubchich et al., 2024).

Two standard formulae have been used for computing the horizontal rate of change of the magnetic field from digitised observatory data. In this study, we examine both of these methods for computing the rate of change and the observed differences that arise between them. The next section describes the differences between the two methodologies, followed by sections on the observed and modeled differences between both. We follow this with a discussion related to scenarios where the results from each method can differ by many orders of magnitude.

## 2  Difference between methods of computing horizontal field change

Representing the horizontal component of the geomagnetic field as a scalar indicator of GIC activity is ambiguous. Consider a digitised time series of orthogonal components of the magnetic field, measured at a permanent magnetic observatory for example. While there is only one correct way to evaluate the first-order time derivative of the magnetic field vector through subtraction of successive readings, several methods can be used to distill the derivative into a scalar value. The scalar magnitude of the horizontal component $B_H$ is defined in terms of the northward and eastward components $B_N$ and $B_E$:

$$B_H = \sqrt{B_N^2 + B_E^2}. \tag{1}$$

One approach to compute the derivative is take the overall magnitude of the horizontal magnetic field $B_H$ and subtract from it the horizontal magnetic field given by the previous measurement, which written in terms of the northwards and eastwards components $B_N$ and $B_E$ when assuming $\frac{\delta B_H}{\delta t} << B_H$, yields:

$$H' = \frac{\delta B_H}{\delta t} = \frac{1}{B_H} \cdot \left( B_N \frac{\delta B_N}{\delta t} + B_E \frac{\delta B_E}{\delta t} \right) = \hat{\mathbf{B}}_H \cdot \frac{\delta \mathbf{B}_H}{\delta t}, \tag{2}$$

where $\hat{\mathbf{B}}$ denotes a unit vector in the direction of the magnetic field and $\delta t$ is the time difference between successive measurements. Alternatively, one can incorporate the rate of change of the magnetic field in the two perpendicular directions $\mathbf{B}_N$ and $\mathbf{B}_E$ separately using their scalar values $B_N$ and $B_E$ by the quantity $R$, given by (Smith et al., 2021):

$$R = |\frac{\delta \mathbf{B}_H}{\delta t}| = \sqrt{\left( \frac{\delta B_N}{\delta t} \right)^2 + \left( \frac{\delta B_E}{\delta t} \right)^2}. \tag{3}$$

This means that $H'$ can be written in terms of $R$:

$$H' = \hat{\mathbf{B}}_H \cdot \frac{\delta \mathbf{B}_H}{\delta t} = |\hat{\mathbf{B}}_H| \cdot |\frac{\delta \mathbf{B}_H}{\delta t}| \cdot \cos\theta = R \cdot \cos\theta \tag{4}$$

where $\theta$ is the difference in angle between the direction of the vector change in the magnetic field between time $(t - \delta t)$ and time $(t)$ and the original magnetic field magnitude at time $(t - \delta t)$. The two methods, $H'$ and $R$, give the same value when the perturbation vector $\delta \mathbf{B} = (\delta B_N, \delta B_E)$ is in the same direction as $\mathbf{B}_H$. One difference between the two is that $R$ is always positive whereas $H'$ can take either sign. However, as space weather applications do not necessarily need to take the sign of GICs into account, this is less important in the comparison between the two methodologies, as a decrease in magnetic field strength is just as effective at inducing a geoelectric field as an increase if they have the same absolute value. The two methods can lead to substantially different results when the magnetic field changes rapidly in direction, especially if this directional change is not associated with a change in magnitude. $R$ will always produce an equal or higher value than $H'$, because in equation (4), $\cos\theta \leq 1$. As a result, $H'$ has a lower magnitude than $R$ unless there is no change in the direction of the magnetic field from one measurement to the next, in which case the two methods produce estimates of the same magnitude (though $H'$ may be negative). In the case where $\cos\theta$ is zero (i.e. $\theta$ is 90°), equation (4) will give a value of zero for $H'$ but $R$ given by equation (3) will be non-zero.

In equation (2), the components $\delta B_N$ and $\delta B_E$ of any perturbation $\delta B_H$ do not have the same weight, as each is multiplied by the magnitude of the field in that direction. For example, if the field is predominantly in the northward direction (as expected for an axial dipole dominated field), then $B_N >> B_E$ and $H'$ is dominated by the perturbation $\delta B_N / \delta t$. As a contrast, in equation (3), the two components are equally weighted. It is worth noting that geomagnetic field data portals such as INTERMAGNET provide direct access to $B_H$ along with the vertical intensity and declination. With $B_H$ directly available from the portal, the use of equation (2) to carry out studies into horizontal geomagnetic field perturbation becomes trivial, whereas to calculate $R$ the declination also has to be used to derive the correct magnitude of $B_N$ and $B_E$.

In addition to considering individual cases when $H'$ and $R$ differ, it is interesting to consider the distribution of $H'$ and $R$ in a large number of trials in each of which the perturbations are randomly drawn. This situation somewhat mimics the geophysical situation where at a given observatory the externally driven field changes may appear quasi-random. In the simplest case, suppose that the perturbations are uniformly distributed in angle and all have the same magnitude. Then $H'$ depends only on the angle between the background field and the perturbation, and because the distribution of the perturbations are rotationally invariant, a histogram of $H'$ will be independent of the direction of the background field, that is, it will look the same at every observatory. Likewise, a histogram of $R$ is independent of background field. Hence, under these assumptions, the distribution of $R$ and $H'$ should be the same everywhere. However, magnetic field perturbations are not uniformly distributed, and there are locations where the background magnetic field is more aligned or less aligned with the externally driven magnetic field perturbations, with this dependence also depending on relative geomagnetic activity (Viljanen et al., 2001). As a result the outputs of the two methods for calculating ground level magnetic field perturbation can in fact be quite different, and will in general depend on location and geomagnetic activity. In the next section, we compute and describe the differences between the two methods using minute-mean data from a set of global observatory measurements. The difference between the methods

could have implications for many studies in space weather research (e.g., Thomson et al. (2011); Mac Manus et al. (2017), both using $H'$, and Smith et al. (2021) using $R$).

Although $R$ and $H'$ are simple measures of horizontal field change, it would be more accurate to model GICs using a convolution integral to incorporate correctly the effect of the finite conductivity of the Earth on GICs, though the conductivity profile is not always available. Viljanen (1998), and Bedrosian (2007), for example, pointed out that the effect of perturbations in the eastwards and northward directions can be treated independently if the geometry of the conductor network (i.e. power grid) impulse response is known. They also noted that the northward magnetic field perturbation dominates GIC activity at stations in Finland for example. The externally driven magnetic field change calculated using the two-dimensional Spherical Elementary Current System (2D SECS) method can also be used as a GIC activity indicator (e.g. Juusola et al., 2023). By including the internal magnetic field component, a dependence on the local conductivity structure of the subsurface would be introduced (Pedersen et al., 2024). This could be a more appropriate indicator of GIC activity than either $H'$ or $R$ if such conductivity data are available.

## 3 Results

### 3.1 Observed difference between $R$ and $H'$

We computed values of both $H'$ and $R$ for 52 geomagnetic observatories from 1998 to 2020 to examine the spatial distribution of differences between these two definitions of horizontal change. The data consist of definitive 1998-2020 INTERMAGNET data (Love and Chulliat, 2013), collected through the VirES server (Smith et al., 2022a, 2025). From this dataset, we took the definitive magnetic field components of $B_N$, $B_E$ and $B_H$ at 1 minute cadence, allowing us to calculate 1 minute resolution magnetic field derivative estimates. Mean monthly sunspot number was downloaded from SILSO (Clette and Lefèvre, 2015). We also utilize the CHAOS-7 magnetic field model (Finlay et al., 2020) to model the effect of declination and intensity on the difference between the two methodologies. We focus on the largest events by only looking at the maximum difference between the methodologies in figure 1, and the largest events constitute the most important features on figures 2 and 3.

Figure 1 shows the absolute difference between $R$ and $|H'|$ at 50 observatories, with two observatories omitted due to anomalously high differences (more than 5,000 nT/min, Narsarsuaq (NAQ, 61.17°N, 45.43°W) and Alibag (ABG, 18.64°N, 72.87°E)). We found a clear dependence on latitude for the relationship between $R$ and $H'$. This is because there is a linear dependence on perturbation size, as from equation (4), $R - H' = R(1 - \cos\theta)$, and average perturbation magnitude increases proportionally to the background horizontal field strength, becoming higher closer to the poles. Three of observatories were chosen for closer investigation as case studies at a variety of latitudes, namely Chambon-la-Forêt, France (CLF, 48.03°N 2.26°E), Tamanrasset, Algeria (TAM, 22.79°N 5.53°E), and Scott Base, Antarctica (SBA, 77.83°S 166.67°E). The statistical properties of $|H'|$ and $R$ for each of these observatories are displayed in table 1. The largest differences between the methods occur during storm times, which are more common during solar maxima. This includes a maximum difference of 707 nT/min for SBA on 29 October 2003.

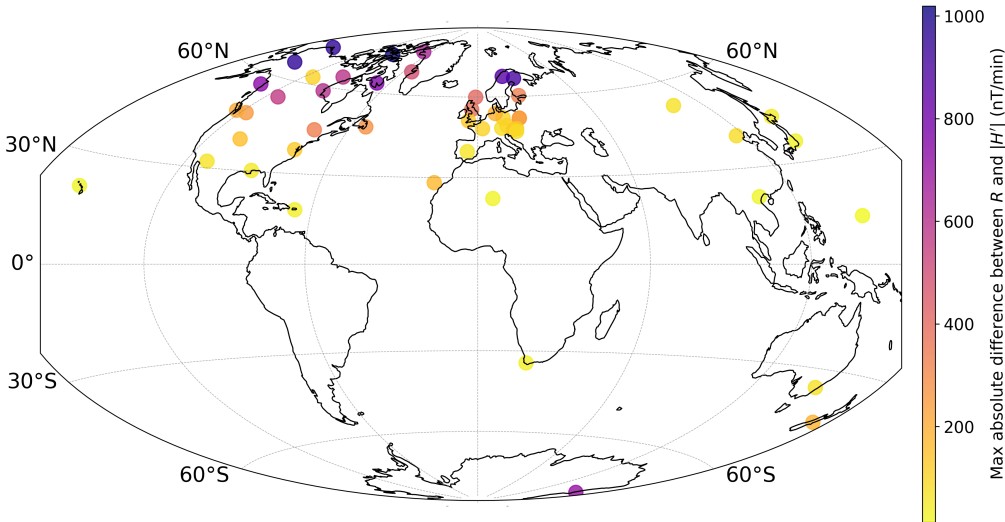

**Figure 1.** The maximum absolute difference between $R$ and $|H'|$ at INTERMAGNET observatories from 1998-2020, found by calculating the difference between the two methods at every minute and plotting the greatest absolute difference.

**Table 1.** Statistics of three stations at different latitudes, with data available from 1998-2020. PCC is the Pearson Correlation Coefficient between $R$ and $|H'|$ over the full period from 1998-2020, and is shown with the mean values of $|H'|$ and $R$ over the same period.

| Observatory | Max difference between $R, |H'|$ | Date of max difference | PCC | Mean $|H'|$ | Mean $R$ | Mean $\frac{R}{|H'|}$ |
|---|---|---|---|---|---|---|
| TAM | 29 nT/min | 5:37 UTC, 31/10/2003 | 0.93 | 0.27 nT/min | 0.38 nT/min | 1.42 |
| CLF | 107 nT/min | 6:59 UTC 29/10/2003 | 0.83 | 0.37 nT/min | 0.66 nT/min | 1.76 |
| SBA | 707 nT/min | 6:25 UTC, 29/10/2003 | 0.79 | 2.31 nT/min | 4.18 nT/min | 1.80 |

We focus on CLF as a mid-latitude observatory that will have the broadest applicability for the United Kingdom and other countries at subauroral locations. A scatter plot between $|H'|$ and $R$ at CLF is shown in Figure 2, with a Pearson Correlation

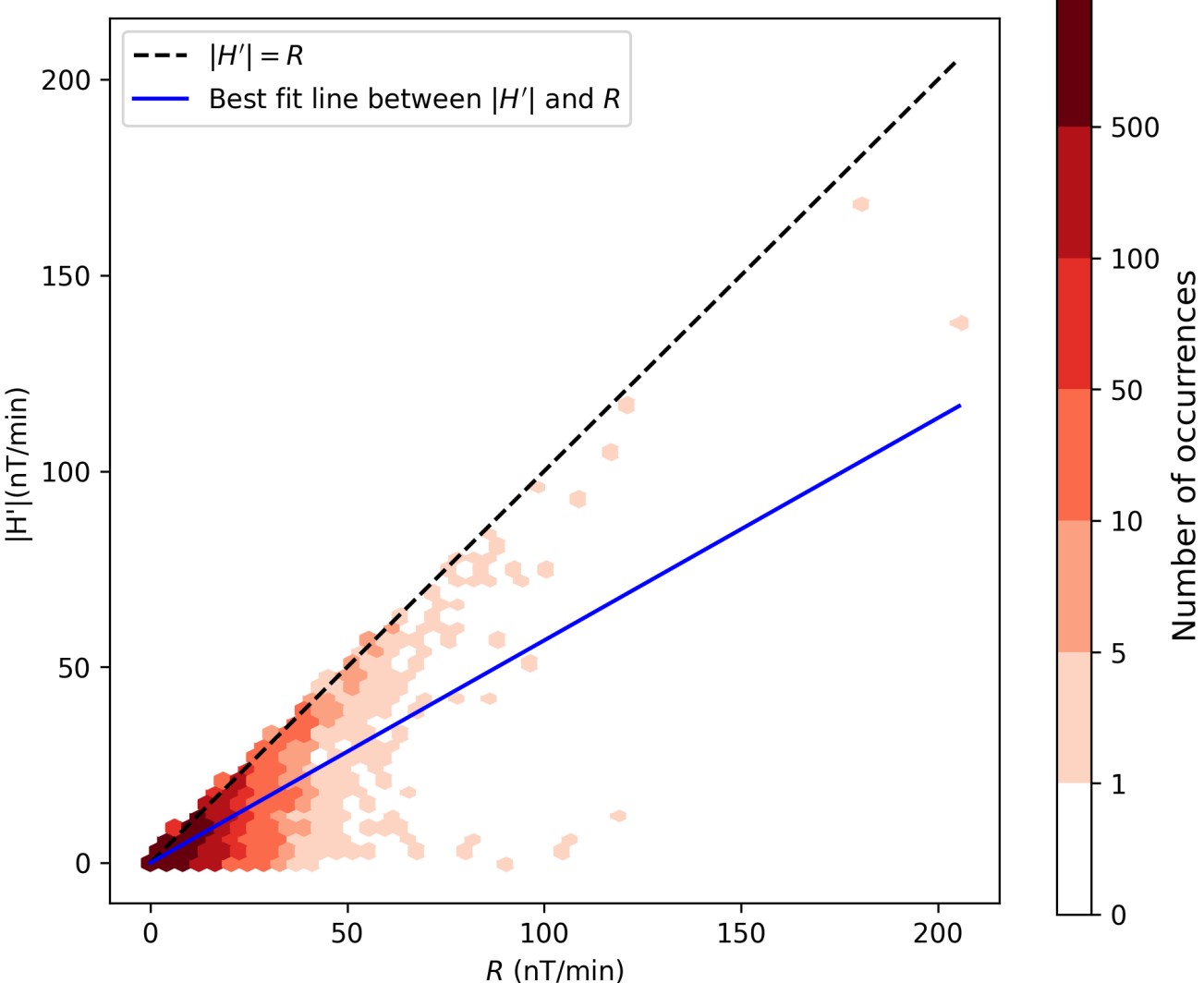

**Figure 2.** Density plot between $|H'|$ and $R$ at CLF from 1998 to 2020. The best fit line intercepts the origin, and shows the line where $R$ is 1.76 times larger than $|H'|$.

Coefficient between the two quantities of 0.83. $H'$ is always less than $R$, and in particular, large $R$ can occur frequently while $H'$ remains low, which is consistent with equation (4). Figure 3 shows the absolute difference between $|H'|$ and $R$ at CLF between 1998 and 2020 along with the sunspot number over the same period. Solar maxima correspond to the highest difference between the methodologies, while solar minima (i.e. 2007-2011) consistently are contemporaneous with low values between the methodologies less than 20 nT/min. In the following sections, we compare the two methods with available GIC

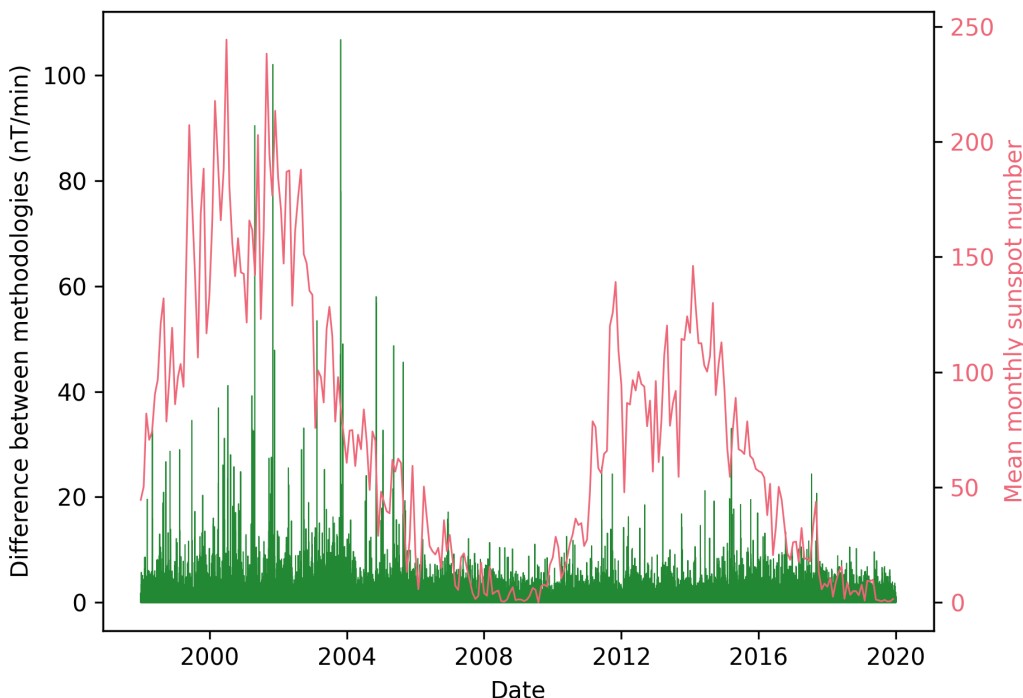

**Figure 3.** Absolute difference between $|H'|$ and $R$ for each minute at CLF from 1998-2020 (green) and the mean monthly sunspot number as an indicator of solar activity (pink).

data and we model the difference between the two quantities $R$ and $H'$ for a given perturbation depending on the declination and inclination of the magnetic field model.

### 3.2 Comparison with GIC data

Rodger et al. (2017) provides a small sample size of 25 events recorded at Eyrewell, New Zealand (EYR, 43.47°S 172.39°E) from 2001 to 2015 and the associated GIC activity at the nearby Islington substation (43.54°S, 172.51°E). The data include

the maximum value of $|H'|$ recorded at that station on a given day, and consist of all dates where that maximum value was greater than 40 nT/min and there is available GIC data at Islington. We verified that the Pearson correlation coefficient between $|H'|$ and GIC was 0.845 and calculated $R$ at those times along with the associated correlation coefficient between $R$ and GIC activity at Islington. The full data table is available in table 2.

We find that the correlation for this sample slightly favours $H'$ as a GIC activity indicator, with a correlation coefficient of

0.830 being calculated for $R$. We found a high correlation between the two quantities $|H'|$ and $R$ with only one data point

| Time (UT) | $|H'|$ (nT/min) (Rodger et al., 2017) | $R$ (nT/min) | ISL M6 GIC (A) (Rodger et al., 2017) |
|---|---|---|---|
| 06/11/2001 01:52 | 190.8 | 190.9 | 33.1 |
| 31/10/2003 05:36 | 170.6 | 172.9 | 21.1 |
| 29/10/2003 06:11 | 166.2 | 166.9 | 34.1 |
| 18/02/2003 05:08 | 109.4 | 109.9 | 12.5 |
| 15/05/2005 08:17 | 97.9 | 97.9 | 15.0 |
| 08/11/2004 07:12 | 90.2 | 127.8 | 14.9 |
| 29/05/2003 22:09 | 88.7 | 89.2 | 14.3 |
| 02/10/2013 01:56 | 85.6 | 86.9 | 19.1 |
| 20/11/2003 18:36 | 78.0 | 81.2 | 12.1 |
| 10/11/2004 02:42 | 74.9 | 75.1 | 14.2 |
| 04/11/2003 06:27 | 73.9 | 74.9 | 13.0 |
| 17/03/2015 04:46 | 68.6 | 68.7 | 17.0 |
| 23/04/2002 04:49 | 66.9 | 67.4 | 11.1 |
| 11/09/2005 05:37 | 62.6 | 62.6 | 19.2 |
| 21/01/2005 23:18 | 60.3 | 60.3 | 8.0 |
| 26/07/2004 22:50 | 57.6 | 63.4 | 18.6 |
| 05/12/2004 07:47 | 56.8 | 56.8 | 8.6 |
| 17/03/2013 06:01 | 54.3 | 55.8 | 13.2 |
| 24/10/2003 15:25 | 52.5 | 55.0 | 7.3 |
| 22/06/2015 18:34 | 51.2 | 52.8 | 12.2 |
| 17/04/2002 11:07 | 47.8 | 49.0 | 6.5 |
| 12/09/2014 15:55 | 43.3 | 47.2 | 9.8 |
| 09/05/2003 07:43 | 42.7 | 42.7 | 4.6 |
| 18/03/2002 13:23 | 41.6 | 41.8 | 5.9 |
| 26/09/2011 19:38 | 40.6 | 40.6 | 8.7 |
| Pearson Correlation Coefficient | 0.845 | 0.830 | |
| $r^2$ | 0.713 | 0.689 | |

**Table 2.** Comparison of $R$ and $|H'|$ at EYR and GIC measurements at Islington substation.

where the two differ significantly, with that being the 8 November 2004 event. This event consisted of a decrease in $B_N$ during an increase in $B_E$, leading to a higher value of $R$ relative to $H'$.

### 3.3 Modeled difference between between $R$ and $H'$

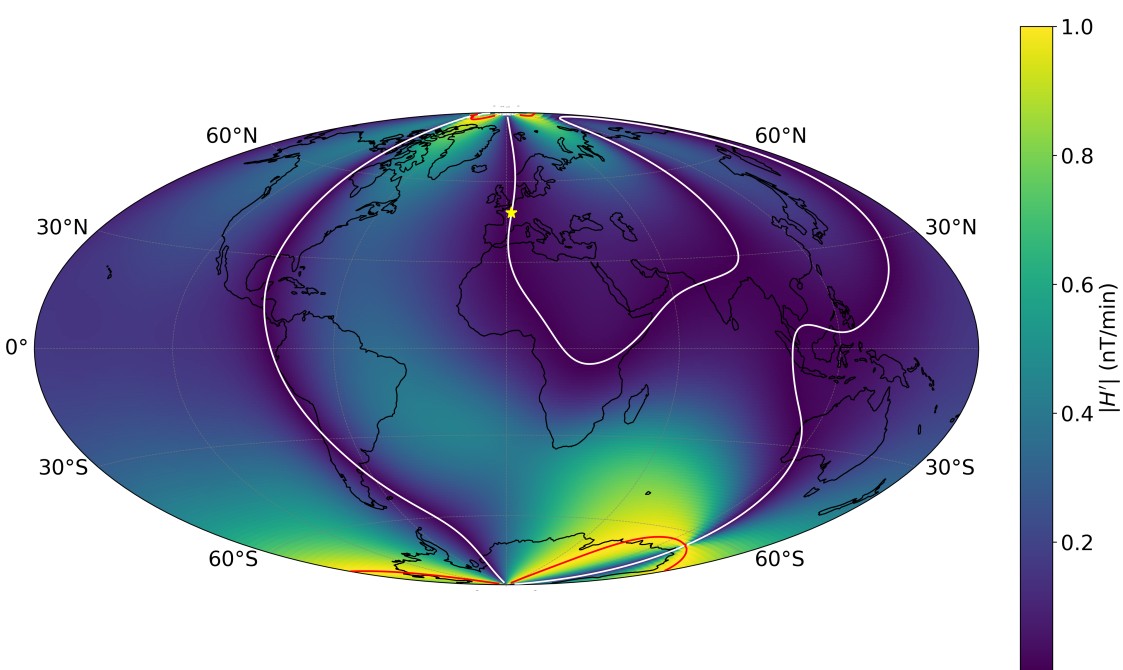

**Figure 4.** The predicted value of $|H'|$ for a perturbation of 1 nT/min in an eastward direction using the CHAOS-7 magnetic field model (Finlay et al., 2020) on 1 July 2013 at 00:00 UTC. The white line shows the agonic line and the red lines mark $\pm 90°$ isogonic lines. The yellow star marks the observatory at Chambon-la-Forêt (CLF), France.

To demonstrate the difference between the two methods, we construct a model which estimates both $H'$ and $R$ using a background magnetic field model. At a chosen epoch, we calculate $B_N$ and $B_E$ from the CHAOS-7 magnetic field model (Finlay et al., 2020). We then add a spatially-constant perturbation of the geomagnetic field in the northward and eastward directions $\frac{\delta B_N}{\delta t}$ and $\frac{\delta B_E}{\delta t}$ and use equations (2) and (3) to calculate $H'$ and $R$ on a $1° \times 1°$ resolution grid over all latitudes and longitudes. As $R$ does not depend on $B_N$ and $B_E$, but only on their time derivatives, $R$ is the same everywhere; however, $H'$ displays significant variability.

Figure 4 shows the result of applying a perturbation of 1 nT/min in an eastward direction on 1 July 2013 at 00:00 UTC, with no perturbation in the northward direction. $R$ is 1 nT/min at every location. In this scenario, as the perturbation is eastwards, a strong northward magnetic field will experience the perturbation as a rotation rather than a change in magnitude, and the eastward contribution to $H'$ will be vanishingly small as in equation (2). As a result, close to the agonic line (where declination is equal to zero, so that the northward magnetic field component dominates), $|H'|$ is significantly smaller than $R$ (coloured dark blue). Conversely, close to the $\pm 90°$ isogonic lines (where declination is $\pm 90°$), $H'$ and $R$ become almost equal (coloured yellow). Figure 4 shows that there can be a significant difference between $H'$ and $R$, even for the same geomagnetic perturbation, which can result in significant differences between the predicted GIC activity levels, because of the varying angle between $\hat{\mathbf{B}}_H$ and $\frac{\delta \mathbf{B}_H}{\delta t}$.

## 4  Discussion

### 4.1  Significance of method choice on peak GIC activity

As observed in the previous section, the largest difference between the methodologies at SBA in the time period measured was 707 nT/min, and this difference is typical of high latitude stations. At CLF, the maximum difference was 107nT/min. The effect of such large differences in the rate of change of $B_H$ could be significant for modelling the induced geoelectric current depending on subsurface conductivity. For example, Ingham et al. (2023) inferred an induced geoelectric field in South Island of New Zealand of around 1.5 V/km with a perturbation of 100 nT over an inducing period of 30 seconds. This has the implication that a significantly larger geoelectric field could be predicted when using $R$ compared to $H'$, having direct implications on GIC risk. The greatest difference between $|H'|$ and $R$ took place during the 2003 Halloween storms of 29–30 and 30–31 October 2003 (Tsurutani and Hajra, 2021) for 17 of the total 52 stations we studied, and 47 of the 52 stations record the greatest difference between the methods when the $K_p$ index (Matzka et al., 2021) is greater than 5. Hence the difference between the methods is largest during storm times, which is when the accuracy of a GIC proxy is the most critical, and indeed Tsurutani and Hajra (2021) found that the largest GICs induced in the Mäntsälä gas pipeline from 1999 to 2019 were induced during the Halloween storms. This would therefore imply that studies into extreme geomagnetic activity (e.g. Thomson et al., 2011) may give different results depending on the methodology chosen. Of the five stations that demonstrate the greatest difference when $K_p \leq 5$, four record the greatest difference at midnight UTC possibly due to artifacts in the observatory time series, and the remaining station is Budkov, Czech Republic (BDV, 49.08°N, 14.02°E), recording the highest difference during a disturbance at 12:09 UTC on 1 March 2016. Differences in the two methods are compounded by rapid changes in the declination at storm times (Rastogi, 2005), which contribute to $R$ but not $H'$.

We find that the publicly available data are not sufficient to come to a strong conclusion from direct comparison with GIC data, with only 25 data points. As GIC activity is heavily reliant on local conductivity structure and conductive network topology(Hübert et al., 2025), we would also hesitate to come to a universal conclusion based on results for a single location and context. We would anticipate a larger study with a sufficient location range within the dataset to make robust conclusions.

Most changes in the auroral electrojet induce northward geomagnetic disturbances (Pirjola, 1998), whereas sudden storm commencements are likely to also produce significant eastward geomagnetic disturbances (Smith et al., 2022b).The ring current contributes to region 2 field aligned currents (Ganushkina et al., 2017), and eastward geomagnetic disturbances have been shown to occur on the dawn side of the worldwide current system at the boundary between the region 1 and region 2 field aligned currents (Bower et al., 2024), suggesting a possible link between the ring current and eastward disturbances at auroral and subauroral latitudes. The equatorial electrojet can induce geomagnetic perturbations in both the northward and eastward directions (Kouassi et al., 2021). These results suggest that $H'$ and $R$ may vary in their effectiveness as a GIC indicator depending on the regime within the nearby ionospheric current system, as an eastward geomagnetic disturbance is likely to produce a larger contribution to $R$ compared to $H'$.

## 4.2   Resolution of observatory data and differences between methods close to the agonic line

The resolution of magnetic field data in the northward and eastward directions at any INTERMAGNET observatory is 0.1nT using a vector magnetometer (Bracke, S. (Ed.) and INTERMAGNET Operations Committee and Executive Council, 2025). This is sufficient for measurements of geomagnetic field perturbation during storm times, where the perturbations are orders of magnitude larger than the resolution. This resolution also does not have a significant impact on the efficacy of either $H'$ or $R$ as indicators of important GICs which would require mitigation or pose a threat to power networks. However, we find that a 0.1nT measurement resolution in combination with the difference between the methods described in the previous section leads to times where $R$ is many orders of magnitude larger than $|H'|$, in contrast to the typical ratio (1.4–1.8) derived in section 3.1. We therefore comment on this artifact within the data and explain how this discrepancy arises.

During quiet times, measurement resolution is of a similar size to the average geomagnetic field perturbation at 1 minute cadence. Analogue-to-digital conversion causes the data to be quantized in certain values (e.g. Bolic, 2023; Colagrossi et al., 2023). The resolution of magnetic field data has an effect on the directional distribution of the observed geomagnetic field perturbation after analogue to digital conversion, as there are only a finite number of options for a vector measurement on a magnetometer with a certain resolution. This has an effect on the difference between $H'$ and $R$, as the size of $H'$ is directly affected by the measured magnetic field direction (equation (2)).

We illustrate the effect of the resolution of a geomagnetic observation on the directional distribution of the measured vector magnetic field perturbation. We model the magnetic field as a background field $\mathbf{B_0}$ plus some perturbation vector $\mathbf{P}$. We assume that the background field does not change with time, such that $\mathbf{B_0}$ is arbitrary and has no effect on the perturbation. We then assume that the perturbation has a fixed magnitude and a uniformly-distributed direction. The objective is then to find the measured direction of $\mathbf{P}$ following analogue-digital conversion. We define the perturbation magnitude as $P$ and the resolution as $\Delta B$.

We provide an example, with $P = 2$ and $\Delta B = 1$ in both the north and east directions. In this case, two perturbations at declinations of –10° and 12° from geographic north will produce measured perturbations with northward components of $2\cos(10°)$ and $2\cos(12°)$, with both perturbations producing a measurement of (2,0) in the (north, east) directions after analogue-digital conversion. An angle of 20°, however, will produce a perturbation of $2\cos(20°)$ and $2\sin(20°)$ in the northward and eastward

directions, with the digital measurements being rounded to (2,1). This produces a measured perturbation at an angle of 26.6°and a magnitude of $\sqrt{5}$. Perturbation magnitudes can also change significantly during analogue to digital conversion, as illustrated in this example.

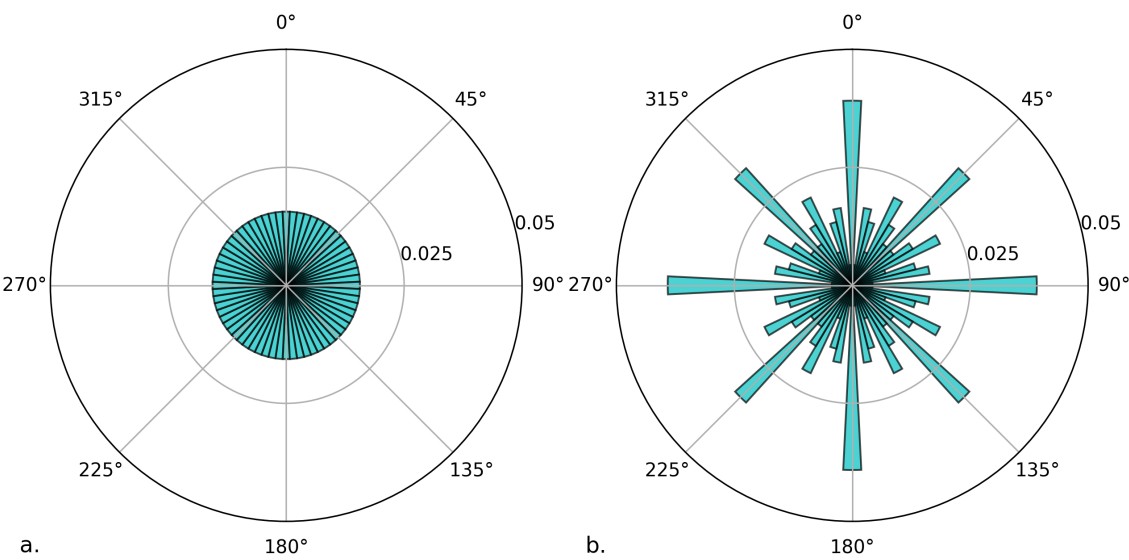

**Figure 5.** Effect of digitisation. Polar plot of perturbation of the analogue (a.) and digitised (b.) magnetic field when the resolution of the recording equipment is 0.1 nT and the original magnetic field perturbation distribution follows a 2D multivariate (bivariate) normal distribution with the covariance matrix 0.146 I, where I is the identity matrix. This is the covariance matrix for a uniform directional normal distribution with a median of 0.45nT/min (the median magnitude of geomagnetic field perturbation at CLF). The radial axis displays the probability of a certain perturbation being within a given angular bin.

In Figure 5, we model perturbations as normally distributed around zero in the northward and eastward directions to produce a illustration of the effect of analogue-digital conversion with as few assumptions as possible. Such a normal distribution is uniform about the mean and has rotational symmetry, which means that there is a uniform directional distribution of perturbation probabilities. This distribution is naïve, as the directional distribution of perturbations are not typically uniform, and the normal distribution will not be representative, especially in active times. However, the distribution allows us to illustrate the effect of digitisation in a controlled fashion.

We take the median perturbation magnitude as 0.45 nT/min, which was the value found for CLF. This was then used to find the corresponding covariance matrix for a bivariate normal distribution with rotational symmetry, which is the identity matrix multiplied by a factor of $\sigma^2$. This was found using the Chi-squared distribution with two degrees of freedom (i.e. the Rayleigh distribution), for which the median $m = \sigma\sqrt{2\ln(2)}$, rearranging to give $\sigma^2 = 0.146$ nT/min. We note that San Juan, Puerto

Rico (SJG, 18.11°N, 66.15°W) had the lowest median perturbation of any station from 1998-2020 at 0.2 nT/min, producing a potentially even more significant effect in analogue to digital conversion. We then find the measured direction after analogue-digital conversion. In Figure 5 we plot the probability of a given perturbation falling within each of 64 bins. The plot shows a significant change in the directional distribution favoring directions that are along the axes of measurement (either directly northward or eastward) due to the points (0,1), (0,2) and others. There is also an increase in frequency at 45° due to the point (1,1). The angular width of each bin is 5.625°. We would expect this quantization to also affect vertical magnetic field perturbations as well.

The 0.1 nT resolution of the data changes the directional distribution of digitised magnetic field perturbations compared to those that actually occurred when the resolution is similar to the median magnitude of the geomagnetic perturbation at a given observatory. Perturbations recorded directly in the eastward (for example) directions occur regularly (Figure 5), leading to the recorded magnetic field perturbation in the northward direction during quiet time being measured as 0 nT/min. Close to the agonic line, a perturbation in the eastward direction will have a vanishingly small contribution to $H'$ because the overall magnetic field vector is perpendicular to the magnetic field perturbation (as in equation (2)). This is also true for northward perturbations close to the $\pm$ 90° isogonic lines. An eastwards perturbation close to the agonic line leads to a value of $H'$ which is orders of magnitude smaller than $R$, demonstrated in Figure 4 with a modeled eastward magnetic field of 1 nT/min. As a result, stations near the agonic line may repeatedly have measurements with $R >> H'$. The agonic line passed through CLF around mid-2013 giving an opportunity to investigate the effect. The times when the recorded perturbation is directly eastward are highlighted by large values of the ratio $Q$ between $R$ and $|H'|$. $Q$ was calculated for each measurement (one per minute), and its rolling average was taken over one week (i.e., 10,080 points at 1 minute cadence) for the full dataset. This allows us to focus on long-term variation. Following from equation (4), $Q = \frac{1}{|cos(\theta)|}$ for the case where $\frac{\delta B_H}{\delta t} << B_H$, such that the rolling average $Q_{RA}(\tau) = \frac{1}{n}\sum_{t=\tau-n}^{t=\tau} \frac{1}{|cos(\theta(t))|}$, where $n$ is the number of minutes of data included in the rolling average. The peak in $Q$ shown in Figure 6 corresponds with $B_E$ approaching zero around the period when the agonic line moves westward (Thompson, 1990) through CLF in mid- to late-2013. The value of $Q_{RA}$ greatly exceeds 10 from 2008-2020 and 100 in 2013. Similar results were found at other stations close to the agonic line, including Stennis Space Center, Bay St. Louis, United States (BSL, 30.35°N, 89.64°W) and Fort Churchill, Canada (FCC, 58.76°N, 94.09°W).

Using the directional distribution of magnetic field perturbations described in Figure 5, we can make a prediction of $Q$ as a function of declination for an observatory with a similar total horizontal field intensity as CLF (40,000 nT). We use a bivariate normal distribution to simulate the magnetic field perturbation distribution, with covariate matrix $0.146\mathbf{I}$ as previously noted. Predicted mean $Q$ over the full digitised distribution for a range of declinations is shown in Figure 7, and illustrates that an increase in $Q$, due to the limited resolution of the file format, is expected around the agonic line, and that other smaller local maxima in $Q$ also exist at other declinations. This corroborates well with Figure 6. The larger range of $Q$ variations in Figure 7 compared with figure 6 is because a bivariate normal distribution is unlikely to be an accurate model of the geomagnetic field perturbation.

The dominant contributions to large values of $Q$ when the declination is small are measurements where $\frac{\delta B_N}{\delta t}$ vanishes (at the recorded resolution of the observations) and when $\frac{\delta B_E}{\delta t}$ is nonzero. An example of this behaviour is shown in Figure 8,

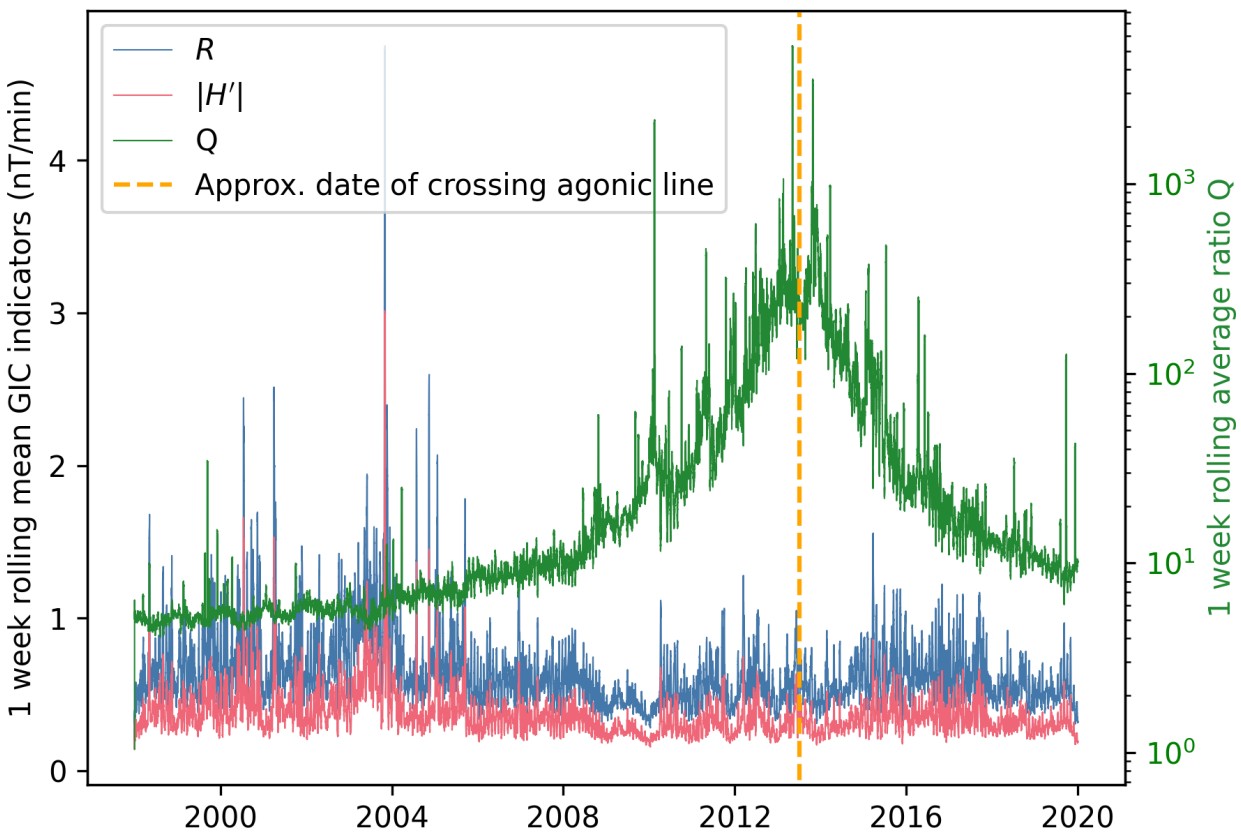

**Figure 6.** The one-week rolling average (10,080 minutes) of the ratio $Q$ for CLF 1998-2020 (green) alongside the one-week rolling averages of $|H'|$ (pink) and $R$ (blue). The approximate date the agonic line passed through CLF is in orange (July 2013).

which is consistent with the hypothesis that high values of $Q$ result from fixed resolution in the file format and proximity to the agonic line. Measurements with the greatest value of $Q$ most commonly coincide with $\frac{\delta B_E}{\delta t}$ also being small (but nonzero), due to smaller perturbations being both more frequent and more influenced by the measurement resolution. Consequently, these intervals do not have a significant contribution to the average absolute difference between $|H'|$ and $R$. Again, this is the case for other observatories we investigated close to the agonic line.

To indicate the potential benefits of improving measurement resolution, we calculated the mean deviation between the observed and digitised declination of the magnetic field perturbation as a function of analogue to digital resolution for the bivariate normal distribution with again a covariance matrix of $0.146\mathbf{I}$ as used previously. We found that a resolution of 0.1 nT

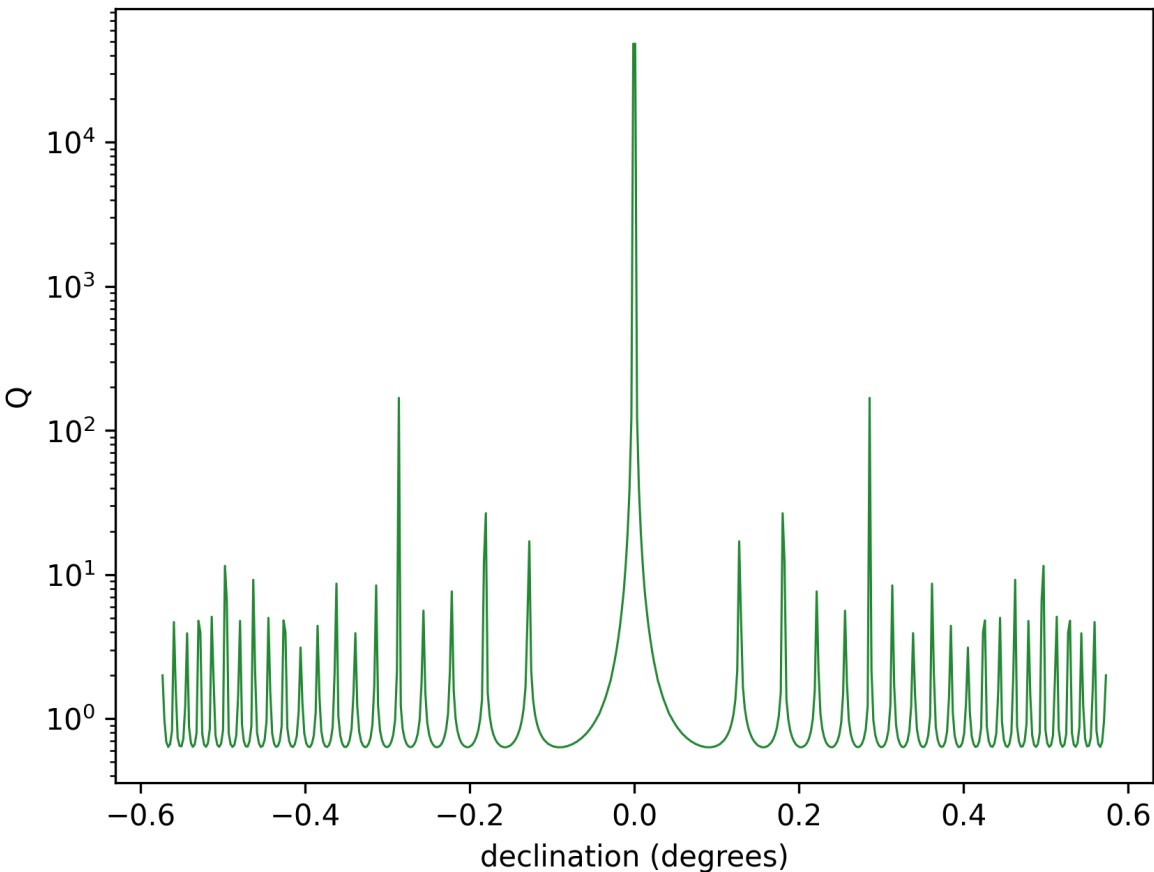

**Figure 7.** $Q$ for an observatory with a magnetic field intensity of 40,000 nT for a range of declinations around the agonic line assuming the bivariate normal distribution of magnetic field perturbations used to produce Figure 5.

(shown in Figure 5) leads to a mean deviation in the measurement of declination of 4.12°. These mean deviations would be reduced to 0.45° and 0.01° for resolutions of 0.01 nT and 0.001 nT respectively, both of which would greatly improve the representation of the observed magnetic field data and reduce the difference between $|H'|$ and $R$ near the agonic line during geomagnetically quiet times.

This investigation implies that the current resolution is having an observable impact on the recorded directional distribution of geomagnetic field perturbations during quiet times, and there are a significant number of instances where no geomagnetic perturbation is recorded in either the northward or eastward directions. We recommend improving the resolution of measure-

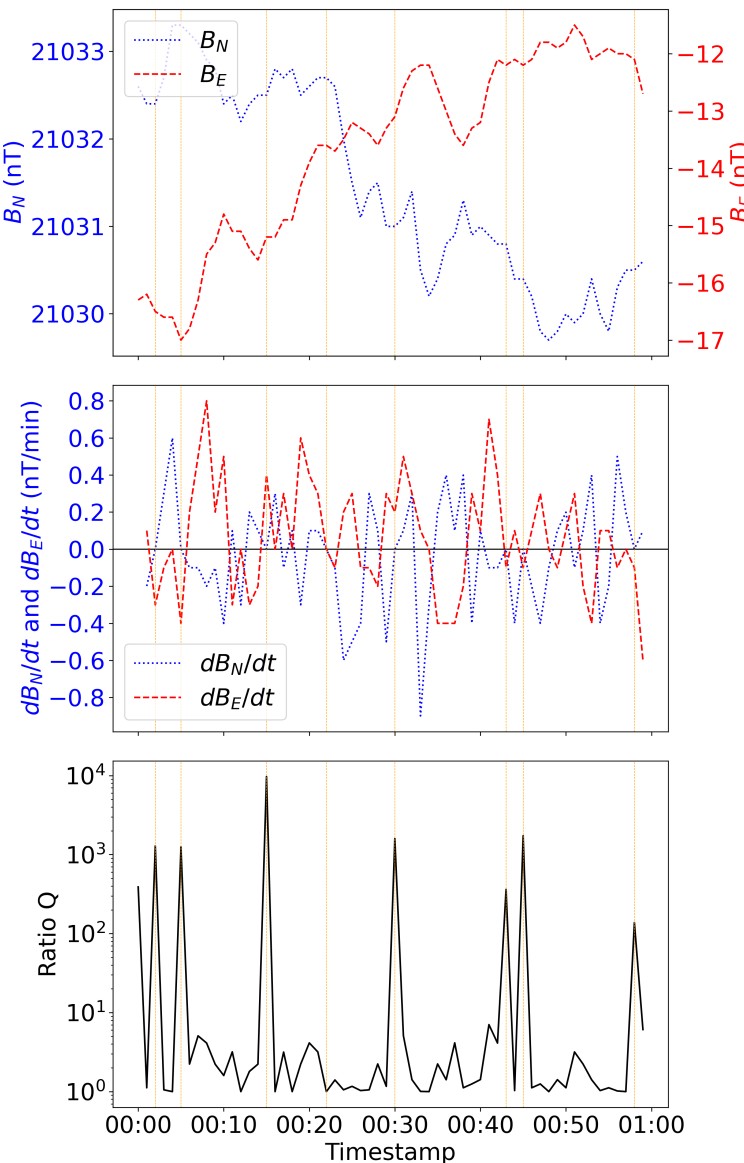

**Figure 8.** Magnetic field magnitudes $B_E$ and $B_N$, their derivatives, and the ratio $Q$ between $|H'|$ and $R$ at CLF for 1 August 2013 between 00:00 UTC and 01:00 UTC. Vertical lines indicate times where $\delta B_N/\delta t$ is equal to zero.

ments of the geomagnetic field to 0.01nT to mitigate this artifact. This is feasible with current fluxgate magnetometer systems
able to reach a precision of around 10 pT (Bennett et al., 2021).

## 4.3 Removal of baseline geomagnetic field

The baseline-subtracted horizontal variation field vector $\Delta\mathbf{B}_H$ is a quantity occasionally used to isolate the externally-driven magnetic field and therefore highlight rapid changes, for example in machine learning research (e.g. Madsen et al., 2022). In these studies, the quiet time background magnetic field is removed so that a regression model can predict solely the externally driven part of the measured variation, simplifying the relationship between inputs (i.e. solar wind parameters) and outputs (i.e. rapid variation in ground level magnetic field) to produce a more explainable and more easily trained statistical model. It might be expected that removing the baseline magnetic field before calculating $H'$ has the potential to remove the effect of the relative magnitudes of different magnetic field directions in equation (3). However, we find that subtraction of the baseline geomagnetic field does not qualitatively change the difference between $R$ and $H'$.

Baseline field subtraction does not change the value of $R$ because it does not depend on the background field. For $H'$ however, the situation is different, and in fact the removal of the baseline can occur either before or after $B_H$ is calculated from $B_E$ and $B_N$ with different effects in each case. Removal of the baseline magnetic field after $H'$ has been calculated introduces an additional term to equation (2) for the rate of change of the magnetic field model. However, this term is vanishingly small compared to the rate of change of the observations, leading to an equation that follows equation (2). This method therefore does not eliminate the difference between $H'$ and $R$. Alternatively, the baseline magnetic field can be removed before combining $B_E$ and $B_N$ to produce $B_H$, producing the scalar magnitude of the baseline-subtracted horizontal variation field vector $\Delta B_H$. Following the subtraction of the previous quantity $\Delta B_H(t-\delta t)$ from this quantity to estimate $\frac{\delta(\Delta B_H)}{\delta t}$, the relative contribution of the horizontal magnetic field perturbation in a given direction is now proportional to the magnitude of the baseline subtracted magnetic field vector $\Delta\mathbf{B}_H$ in that direction since

$$\frac{\delta(\Delta B_H)}{\delta t} = \frac{1}{\Delta B_H} \cdot (\Delta B_N \frac{\delta(\Delta B_N)}{\delta t} + \Delta B_E \frac{\delta(\Delta B_E)}{\delta t}) = \Delta\hat{\mathbf{B}}_H \cdot \frac{\delta}{\delta t}(\Delta\mathbf{B}_H) \tag{5}$$

where $\Delta B_N$ and $\Delta B_E$ are the northward and eastward components respectively of $\Delta\mathbf{B}_H$. As $\Delta\mathbf{B}_H$ does not have a symmetric directional distribution and has a significantly different distribution to any rapid geomagnetic field perturbation depending on location (Viljanen et al., 2001), removal of the baseline magnetic field does not remove the difference between $H'$ and $R$. A large geomagnetic perturbation nearly perpendicular to $\Delta\mathbf{B}_H$ will still cause relatively little change in the overall magnitude of $\Delta\mathbf{B}_H$, and will therefore lead to a large discrepancy between $H'$ and $R$. This means that a machine learning model that uses a baseline subtracted magnetic field variation vector as the target for its training and testing, for example as its GIC indicator, will still give different results depending on which of the two different methods was used to calculate the horizontal magnetic field perturbation.

## 5 Conclusions

There are two common methods, denoted $H'$ and $R$, for computing the rate of change of the horizontal magnetic field. $H'$ is calculated by subtracting successive scalar magnitudes of $\mathbf{B}_H$, while $R$ is calculated from the difference between two

successive vector measurements of $\mathbf{B}_H$. We show that significant relative and absolute differences can arise between them, particularly during storm times. As a result, calculations of ground level magnetic field perturbations should consider this factor carefully if they are intended as indicators of GIC activity.

We also investigated the relative ratio of $H'$ and $R$ for observatories close to the agonic line. We find that the fixed format of minute mean data to one decimal place can significantly affect the directional distribution of recorded magnetic field perturbations, and has an effect on the computation of $H'$, though this effect is not significant during storm times when GIC activity indicators are the most critical. We recommend improving the resolution of recorded magnetic field data to 0.01nT or better, which will facilitate studies into the directional distribution of magnetic field perturbation.

*Data availability.* INTERMAGNET data are accessible at https://intermagnet.org/data_download.html, containing all data used for the analyses within this manuscript.

*Author contributions.* SAF conceptualized the study, made the present analyses and wrote the manuscript with the contribution of the other authors. PL, CB, KW and GR contributed directly to the writing of the manuscript through reviews and edits, were responsible for the structure and style of the manuscript, and assisted with the submission process. PL contributed to the interpretation of the results, contributed
to the structure and message of the paper, and identified the reason for the difference between the two methodologies close to the agonic line. KW supervised the study along with GR, CB and PL.

*Competing interests.* The authors declare that they have no conflict of interests

*Acknowledgements.* Funding for this research was provided by NERC through a SENSE CDT studentship (NE/T00939X/1). Additional funding was received from the British Geological Survey (BUFI Number S477) and the United Kingdom Meteorological Office. We thank
Craig Rodger and the Solar Tsunamis team at the University of Otago and their collaborators at Transpower for the GIC data used in this study. The results presented in this paper rely on data collected at magnetic observatories. We thank the national institutes that support them and INTERMAGNET for promoting high standards of magnetic observatory practice. (https://www.intermagnet.org/)

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
