# Peer review of "A comparison of methods to compute the rate of horizontal geomagnetic field variation"

_EGUsphere, 2025_

## Author Response (AR1)

**RESPONSE TO COMMENTS ~ A comparison of methods to compute the rate of horizontal geomagnetic field variation**

**Please note that the line numbers in this author response refer to the line numbers on the original manuscript rather than any updated manuscript.**

**RC1:** 'Comment on egusphere-2025-2857', Anonymous Referee #1, 28 Jul 2025
Review comments on the manuscript egusphere-2025-2857 entitled "A comparison of methods to compute the rate of horizontal geomagnetic field variation" by Fielding et al.

This study deal with possible indicators for geomagnetically induced currents (GIC). The authors have presented a comparison between two approaches that determine change rates of local magnetic variations. Several authors have used the temporal changes of the horizontal field component, H', as an indicator for GIC activity. Others have used the modulus of the change rates from the two horizontal components, R, as the indicator. As expected, R is always larger than H', and this difference depends on the local declination angle. The statistical characteristics of the R/H' ratio are investigated and also its dependence on the resolution of the recordings.

While the presented results clearly show the advantage of R over H' time series, the study is limited to technical descriptions. It provides next to no scientific interpretation. I could imagine that such reports are more suitable for other journals. Readers of Annales Geophysicae expect more scientifically relevant articles.

**Response:**

The study takes both an empirical and theoretical look at the difference between the two methods using observatory data and synthetic data, finding that the difference is significant in certain situations, and providing evidence. The study is the first to explicitly note the effect of quantization due to analogue to digital conversion and how this affects the direction of the measured magnetic field. The article provides a starting point for the discussion on which methodology is most suitable for GIC studies. The lack of GIC data is the reason we do not strongly make a recommendation on which of the methods to use at the end of the paper, but point that the two methods can produce significant differences, especially during storm times, if they are being used as a proxy for ground hazards. We agree that a closer look at measured GIC would be a good idea to allow better scientific interpretation, as one of the two indicators R and H' might correspond better with measured GIC. However, there are very few available GIC measurements. In response to this comment, we have added a short section relating to GIC measurements in New Zealand following line . However, we would point out that the response of the geoelectric field to local conductivity changes the response from being purely dependent on the rate of change of H (e.g. Hubert et al, 2025) which further complicates a direct comparison (as noted in the manuscript following line 172).

Further comments are detailed below.

**General comments**

1. Relevant GICs are caused by rapid and large changes of the geomagnetic field. Since this is the focus of the study, it is not very helpful to look at the statistical properties of field variations at a certain station. Your extended elaboration of the situation near 0° Declination and the discussion of magnetometer resolution makes not much sense in the context of GICs. Of relevance are the few percent of largest changes. From the nice and large dataset, I could think of taking just the 1% largest cases at the various stations. Determining from them the R to H' properties, e.g. latitude dependence, local time variation, etc. would make more sense.

**Response:**

It is helpful to look at the statistical properties of field variations during quiet time. Though these statistical properties are not relevant for large events, it needs to be noted when doing a comparison study that R and H' can differ by many orders of magnitude under some circumstances, and the explanation provides a strong secondary conclusion that the resolution of observatory data is not sufficient for studies of directional distributions of perturbations at 1 minute cadence. While we do not look explicitly at the 1% largest cases, the content of Section 3 discusses the largest events; Figure 1 shows the maximum difference between H' and R at a given location, for the strongest events. Figures 2 and 3 also highlight the differences between the methods during the largest events, although we show smaller events too. We have added a comment that, although we have not selected the largest events specifically, they are the focus of our analysis. This comment is inserted on line 121.

1. Your study implies that the R to H' ratio is mainly a determined by the declination angle. If that is correct, R and H' do an equally good job because the declination is rather constant. The conversion to GIC would require just a different scaling factor. However, it is expected that the focusing on the large cases will provide different answers, depending on the latitude range and other factors. Here more scientific aspects could be considered in the study. What are the main current types that cause the severe changes at the latitude ranges? Are these electrojets, affecting mainly the north component, or the travelling westward surge at substorm onset, which are clearer in the east component, effects of SSC and ring current changes, best seen in H component. Such an investigation would give the user a scientifically justified recommendation at hand.

**Response:**

We note that theta in equation (4) is not the declination, but rather the angle between the direction of the vector change in the magnetic field between time (t-dt) and time (t) and the original magnetic field magnitude at time (t-dt). We have updated the sentence in line 70 to clarify. The ratio between R and H' is not only determined by the declination angle, but is influenced by a number of other factors. We found that the ratio between R and H' also changes with the size of the perturbation, for example.

In the Introduction we note (line 25) that "Geomagnetic disturbances are primarily driven by currents in the magnetosphere and ionosphere especially the auroral electrojet at mid to high latitudes (Pulkkinen et al., 2017), though the equatorial electrojet and the ring current can also contribute to field variations at lower latitudes (de Villiers et al., 2017)." To improve this characterisation, we have added some further references and information on the current systems involved as well as the effects that these different current systems may have on the northward and eastward directions of the magnetic field, and by extension the efficacy of H' and R as indicators of GIC activity. We have inserted a section following line 27 which we think satisfies this. However, we would hesitate to make conclusions on justifying a recommendation on which of R or H' to use based on the current type at a given location or time, and we would simply like to conclude that the efficacy of H' or R probably does depend on the current ionospheric current system nearby to the location in question. Despite that, we add a short section in the discussion that refers to which current systems may contribute to the different quantities. This is located following line 172.

**Special comments**

Line 56: Delete "there are"

**Response:**

This change has been made in the text.

Line 60ff: It is not clear how you derived the differences shown in Figure 1. H' is varying between positive and negative values, R is positive definite. In the case of a sine wave H' has the same deflections in positive and negative direction. R from Eq. (3) has only positive values, it comes at twice the frequency, and the peak-to-peak differences are half as large. Please describe in detail how you arrived at the values shown in Figure 1 and the following figures.

**Response:**

We replaced H' with $|H'|$ in the figure caption and within the prose, and on figures 2, 3. In terms of the second part of the comment, "R from Eq. (3) has only positive values, it comes at twice the frequency, and the peak-to-peak differences are half as large", this is true for the special case in which one of either $B_N$ or $B_E$ are sinusoidal oscillations sin (t) and the other is constant, in which case H' becomes a sinusoid but for R the absolute value is taken, leading to a peak-to-peak difference half as large and a frequency of double that of H'. However, once we consider that $|H'|$ (as shown in Figure 1) has the negative values flipped to positive, we find that the hypothetical scenario described above results in the same value for both R and H', with the same frequency and peak-to-peak differences. Hopefully this explanation helps explain the contents of Figure 1 and subsequent figures. We have not included this explanation in the text.

Line 227: Delete "for each"

**Response:**

This change has been made in the text.

**RC2**: 'CC2 again as RC', Bruce Tsurutani, 15 Aug 2025
Review of "A Comparison of Methods to Compute the Rate of Horizontal Geomagnetic Field Variation" by Fielding, Livermore, Beggan, Whaler and Richardson

The authors are to be commended for doing careful analyses, however when reading the paper one is continually asking the question why are only Bn and BE considered? Maybe those are the dominant magnetic components during quiet intervals, but what about magnetic storm times? I think the authors have not kept up with the current literature. I suggest that they read several recent works and rethink their method of analyses. See Geomag. Aer., 2024, 64, 6, 833-844; SW, 21, e2022SW003383, 2023. https://doiorg/10.1029/2022SW003383; Nat. Port. Sci. Rpts, 14:25074, https://doi.org/10.1038/s41598-024-76449-z; J. Atmos. Sol.-Terr. Phys., 261, 2024. It is clear from these papers that the magnetic perturbations can occur quickly and in all directions during magnetic storms. These papers should be referenced and mentioned in the introduction section of the paper.

**Response:**

Thank you to the reviewer for their constructive comments. We address the reason for using $B_N/B_E$ briefly in line 44: "Considering the time derivative of the 2D horizontal vector geomagnetic field rather than the total magnetic field is based on the induction response as used in

magnetotelluric sounding. This establishes a frequency-dependent relationship between the magnetic to electric field variations as measured in the North (BN ) and East (BE ) directions (Robertson, 1987)." Therefore, although the magnetic field does change in the BC direction significantly during storm time, it is regularly not considered important for GIC induction, and there are many papers cited within our manuscript that only look at horizontal components as a GIC indicator [Marshall et al, 2012, Rodger et al, 2017]

Firstly, although we justify the use solely of B_N and B_E, we now refer to the relevant recent works suggested by the reviewer in the introduction (Geomag. Aer., 2024, 64, 6, 833-844; SW, 21, e2022SW003383, 2023. https://doiorg/10.1029/2022SW003383; Nat. Port. Sci. Rpts, 14:25074, https://doi.org/10.1038/s41598-024-76449-z; J. Atmos. Sol.-Terr. Phys., 261, 2024) following line 40, making it clear that the B_C component does change as well on line 46. As a part of this, we also include that GICs can also affect gas pipelines on line 25. We have also added a further citation on line 46 which investigates whether B_C can be geoeffective.

Secondly, it is worth noting that section 4.2 is not specifically related to GIC studies. The analogue to digital conversion will have a similar effect in three dimensions as well as two, leading to quantised values for the vertical rate of change as well as the horizontal, and therefore at the geomagnetic poles where the vertical field component approaches 90 degrees or the geomagnetic equator where the vertical component approaches zero, we should also see the scenario where the equivalent quantities of H' and R in three dimensions will also peak due to digital to analogue conversion. we have therefore given a brief statement within that section which addresses this point (line 167).

In the Introduction Section, the references quoted are no. primary papers.  I suggest that the authors do some basic literature search and come up with some more basic references.

Line 15, probably an early reference to CMEs (and sheaths) is JGR, 93, A8, 8519-8531, 1988.

**Response:**

This change has been made in the text.

Line 16.  Two good references for solar energetic particles are: SSR, 90, 413–491,1999, doi:10.1023/A:1005105831781 and JGRSP 2024 *129*, e2024JA032622. https://doi.org/10.1029/2024JA032622

**Response:**

This change has been made in the text.

Line 18.  Better references are: Phys. Rev. Lett., 6, 47, 1961 and JGR, 77, 16, 2964-2970, 1972,

**Response:**

This change has been made in the text.

Line 19. A better reference for the enhancement of the radiation belts is JGR 76, 16, 3587-3611, 1971.

**Response:**

This change has been made in the text.

Line 20.  There has to be an earlier and better reference for the aurora?

**Response:**

A 2004 reference to the aurora is inserted.

Line 21.  A reference for substorms is needed here.  It is PSS 12(4), 273–282.
https://doi.org/10.1016/0032-0633(64)90151-5

**Response:**

This change has been made in the text.

Line 25.  Add reference JSWSC, 2021, 11, 23, https://doi.org/10.1051/swsc/2021001.  This paper
notes that the biggest GICs at the Mantsala Norway pipeline during a 21 year study were detected
during the Halloween 2003 magnetic storms.  I will come back to this point later.

**Response:**

This change has been made in the text.

Line 41, the time derivative of the magnetic field needs a reference.  See  GRL 2014, 41,
doi:10.1002/2013GL058825.

**Response:**

This change has been made in the text.

Lines 164-166.  There are several "Halloween storms".  Please be specific.  See the reference for line
25.  The two Halloween magnetic storms had the highest GICs in the 21 years of Mantsala pipeline
data.  So what does this tell you about future efforts for finding a proxy for GICs?  What can you do
to improve your model?

**Response:**

This is a valid point; indeed, some of the 47 stations mentioned experienced the greatest difference
during the first storm and some during subsequent. We now clarify that we refer to the Halloween
Storms (with dates) of 2003, with the associated reference JSWSC, 2021, 11, 23,
https://doi.org/10.1051/swsc/2021001. In terms of the second half of the point, we now state this
as a reference in the line 166 "…the methods is largest during storm times, which is when the
accuracy of a GIC proxy is the most critical", with a short statement pointing out that Mantsala found
the highest GIC specifically during this time where 47 of the 52 stations had the highest difference
between the two methods.

In the Introduction you need to inform the readership that the biggest GICs are associated with
magnetic storms.  Since that is the case you need to focus your study on field variations that are
applicable to storm intervals.

**Response:**

I completely agree with the statement that we need to inform the readership that the biggest GICs
are associated with magnetic storms. This was not specifically mentioned in the introduction, and it
should be clearly stated, along with the fact that storm intervals (as a result) are the most important
target for the horizontal magnetic field study. We have therefore included the statement "These

disturbances and their induction of GICs are most significant during strong geomagnetic activity (citation), and the largest GICs are induced during the most active geomagnetic storm times such as the 2003 Halloween storms (citation)" following line 27.